# Combined Approach of GWAS and Phylogenetic Analyses to Identify New Candidate Genes That Participate in *Arabidopsis thaliana* Primary Root Development Using Cellular Measurements and Primary Root Length

**DOI:** 10.3390/plants11223162

**Published:** 2022-11-18

**Authors:** Brenda Anabel López-Ruiz, Elsa H. Quezada-Rodríguez, Alma Piñeyro-Nelson, Hugo Tovar, Berenice García-Ponce, María de la Paz Sánchez, Elena R. Álvarez-Buylla, Adriana Garay-Arroyo

**Affiliations:** 1Laboratorio de Genética Molecular, Desarrollo y Evolución de Plantas, Departamento de Ecología Funcional, Instituto de Ecología, Universidad Nacional Autónoma de México, Ciudad de México 04510, Mexico; 2Departamento de Producción Agrícola y Animal, Universidad Autónoma Metropolitana-Xochimilco, Ciudad de México 04510, Mexico; 3Centro de Ciencias de la Complejidad, Universidad Nacional Autónoma de México, Ciudad de México 04510, Mexico; 4División de Genómica Computacional, Instituto Nacional de Medicina Genómica (INMEGEN), Ciudad de México 14610, Mexico

**Keywords:** *Arabidopsis thaliana*, accessions, primary root growth, meristem zone, elongation zone, GWAS

## Abstract

Genome-wide association studies (GWAS) have allowed the identification of different loci associated with primary root (PR) growth, and Arabidopsis is an excellent model for these studies. The PR length is controlled by cell proliferation, elongation, and differentiation; however, the specific contribution of proliferation and differentiation in the control of PR growth is still poorly studied. To this end, we analyzed 124 accessions and used a GWAS approach to identify potential causal genomic regions related to four traits: PR length, growth rate, cell proliferation and cell differentiation. Twenty-three genes and five statistically significant SNPs were identified. The SNP with the highest score mapped to the fifth exon of *NAC048* and this change makes a missense variant in only 33.3% of the accessions with a large PR, compared with the accessions with a short PR length. Moreover, we detected five more SNPs in this gene and in *NAC3* that allow us to discover closely related accessions according to the phylogenetic tree analysis. We also found that the association between genetic variants among the 18 genes with the highest scores in our GWAS and the phenotypic classes into which we divided our accessions are not straightforward and likely follow historical patterns.

## 1. Introduction

GWAS (Genome-Wide Association Studies) are a powerful tool for identifying genes that participate in complex phenotypes due to the association between a particular phenotype and genetic markers such as single nucleotide polymorphisms (SNPs) [1]. *Arabidopsis thaliana* (from now on Arabidopsis) has been widely used for GWAS due to the availability of many sequenced natural isogenic lines [2,3].

The root is an essential organ that provides plants with anchorage, acquisition of nutrients and water uptake from the soil [4]. The root has become an organ model because it is relatively easy to study and cells transit from a proliferation phase to an elongation and a differentiation state along the longitudinal axis. In addition, it displays a very plastic development depending on the environment in which the plant grows [5,6]. Therefore, the primary root (PR) is an excellent model to study the integration of environmental inputs into developmental decisions of cell proliferation/differentiation [3].

There has been an increased effort when using GWAS to discover the causal mechanisms involved in PR growth in response to a particular stress condition such as salinity [7,8], nutrient deficiencies, nutrient excess [9,10,11,12,13], soil contamination [14] and wounding [15]. Nevertheless, few GWAS studies have focused on discovering new genes involved in PR growth under control conditions [16,17] and we have only found one report that analyzed root traits at the cellular level [18].

The longitudinal organization of the PR, from the tip to the base of the stem, has different domains and zones: at the root tip is the root cap, formed by the columella, and the lateral root cap, followed by the meristematic zone (MZ), the elongation zone (EZ) and the differentiation zone (DZ) [19,20,21,22]. The MZ is where new root cells are produced, and it consists of the Stem Cell Niche (SCN), the Proliferation domain (PD) and the Transition domain (TD). The SCN comprises of the central organizer, known as the Quiescent Centre (QC), which is enclosed by stem cells that self-renew and divide to provide cells to the columella or the PD-TD, where cells duplicate to produce most of the root cell types [6,20,21]. Subsequently, the cells of the MZ transit to the EZ where they exhibit cell wall loosening, endoreduplication and rapid cell expansion [23]. Then, the cells reach the DZ, halt their elongation and acquire their final features, such as the presence of epidermal root hairs [6,23].

The number of cells produced at the MZ and the length of elongated and differentiated cells determine the PR length [6,20,21,24]. It has been reported that both processes can lead to a shorter PR. For example, this has been observed in the loss of function mutants of *XAANTAL1* and *XAANTAL2* MADS-box genes, through the disruption of hormonal homeostasis or by different stress conditions [25,26,27,28,29]. Interestingly, and despite the high correlation found between the meristem length and the length of the completely elongated cells [18], there are several examples where a short PR is generated due to a shorter meristem, without changes in the length of fully elongated cells [30,31,32].

To provide insight into the genes that participate in the control of cell proliferation, cell differentiation and PR growth, we analyzed the natural genetic variation present in populations of Arabidopsis using 124 accessions. We focused on four traits with the potential to provide new and interesting information pertaining to the genetic bases of primary root growth: the PR length kinetics over a 12-day period, starting from day 2 after sowing; the PR growth rate, the length of the meristem and of the length of the elongation zones. In each case, we found phenotypic variation among the accessions that we classified into small, medium and large for three different traits: PR growth, length of the meristem and elongation zone, while the growth rate was divided into slow, medium and fast. As expected, we detected a strong correlation between the PR growth rate and the PR length; also, we found a medium correlation between the meristem and the PR length, and a small, positive correlation between the EZ length with either the MZ length or the PR length.

Afterwards, GWAS were performed to identify associations between the trait variants among the 124 accessions and different SNPs. As a SNP marks a particular genomic region, which is not necessarily a coding or regulatory sequence, we used a 10-kb window and discovered 23 candidate genes mainly associated with PR length. One of the genes that we obtained codifies for a *FATTY ACID DESATURASE 2*, which has been previously shown to be a promoter of PR growth [33]. Interestingly, another gene was NAC domain-containing protein 48 (*NAC048*) that has different SNPs in its exons depending on whether the accessions develop a short or large PR; moreover, we found that the accessions with a short PR which bare an SNP in their third exon are closely related according to the phylogenetic tree analysis.

## 2. Results

### 2.1. Natural Allelic Variation Determines Primary Root Development in Arabidopsis

The 124 accessions were grown in seven different batches with Col-0 as an internal control and batch effect was not detected (Appendix A). We measured the PR length from 2 days post sowing (dps) until day 12, by marking the position of the root tip every 24 h (Figure 1A,B). The variability of the PR root time course growth of the 124 accessions (Appendix A) performed during 12 dps can be observed in Appendix A. In addition, we used 5 dps seedlings (Figure 1A) to perform the different cellular measurements. In this work, the meristematic zone corresponds to the length from the quiescent center (QC; yellow arrowhead in Figure 1C) to the cortical cell before the cell whose size is twice the size of the previous cell (white arrow in Figure 1C), whereas the EZ length was defined as from the end of the meristem zone to the first root hair bulge (white arrowhead in Figure 1C). The 124 accessions were ranked for each trait and the mean and standard deviation are presented in Appendix A.

Furthermore, we classified the accessions according to the standard Sturge’s rule. As expected, we found a continuous distribution (Appendix A) suggesting complex phenotypes controlled by multiple genes. Since we found many different classes in each distribution, we decided to group them into only three classes: short, medium and large, also according to Sturge’s rule, to analyze the phenotypes more accurately (Figure 1D). This classification allowed us to detect contrasting accessions to further analyze for—polymorphisms in the genes obtained by GWAS, as well as associations in the phylogenetic trees (see below Section 2.2 and Section 2.3).

Using only the 12 dps root length measurements, we classified 27 accessions (ranging from 3.14 to 5.25 cm) as short whereas 79 accessions were assigned as medium (from 5.3015 to 7.3141 cm) and 18 accessions as large (from 7.41 to 9.51 cm) (Appendix A, Figure 1D and Figure 2A). In addition, accessions with different meristem lengths were classified as short (125.13–172.53 μm), medium (173–219.63 μm) or large (221.63–268.20 μm), including 31, 78 and 15 accessions in each case, respectively (Figure 1D and Figure 2B). Moreover, we also classified these accessions according to EZ size, obtaining groups of short (329.995–493.604 μm) and medium EZ (498.02–649.42 μm), with nearly the same number of accessions (55 and 56, respectively), whereas only 13 accessions displayed a large EZ (667.59–827.46 μm) (Appendix A, Figure 1D and Figure 2C). Additionally, the growth rate was divided into slow, medium and fast (Appendix A).

For all accessions, 27 (33.3%) with short PR also displayed a short meristem and a short EZ length (GD-1, BA1-2, BAA-1, C24, HI-0, LO-2, LP2-2, LU-1 and MT-0); from these, GD-1 is the accession with the shortest PR length (3.14 ± 0.39 cm; see Appendix A), a short meristem (161.72 ± 25.05 μm) and a short EZ (371.04 ± 93.74) but are not the shortest in the last two traits. Furthermore, 26 accessions (20.96%) have medium PR length, meristem and EZ (Appendix A and Figure 2) but there are others with medium PR length, such as LAC-3 (5.37 ± 0.51 cm) and PHW-2 (6.84 ± 0.416) with short (145.04 ± 17.55) or large (236 ± 19.10) meristem, respectively (Figure 2 and Appendix A). In addition, of the 18 accessions classified as displaying a large PR, 6 of them (33.3%) (CRL-1, HS-0, KA-0, MNZ-0, RA-0 and WIL-1) have large meristems but medium EZ; of these, RA-0 has the largest PR (9.515 ± 0.92 cm) (Figure 2). It is noteworthy that BUR-0 is the only accession with a large PR and a large meristem and EZ. Moreover, we found that 22.2% of accessions with large PRs also have a large EZ (BUR-0, JM-0, LOV-1, RRS-7) (Figure 2).

In addition, of the 27 accessions classified as having a short PR, 15 have a slow growth rate, and the rest (12) have a medium growth rate. Likewise, of the 79 accessions categorized as having a medium PR, 59 have an intermediate growth rate, 15 have a fast growth rate and 5 have a slow growth rate. The accession KA-0, which has a long PR length, has the highest growth rate (Appendix A, Figure 2).

To understand how these cellular parameters and PR length are associated, we performed multiple correlations among the four variables (Figure 3), finding that the highest correlation (0.85) detected was between the PR length at 12 dps and the growth rate that was obtained using the difference in growth between day 12 and 11. Also, we found that the correlation between the PR length at 5 dps and 12 dps is 0.684, showing differences in growth during development. Different to what we expected, both the meristem and the EZ size have a higher significant correlation with the PR length at 12 days than at 5 days, suggesting the importance of considering additional temporal measurements to explain the PR length among different accessions. Moreover, there was a low positive correspondence among the meristem size, the growth rate and the EZ length, in contrast to a negative correlation between the EZ and meristem length previously reported [18] (Figure 3).

### 2.2. GWAS Reveal Genes That May Explain the Natural Variation in the Length of the Primary Root, the Meristem and the Elongation Zone

We performed independent GWAS using measurements from the 124 accessions for PR course growth (2–12 dps), meristem size, EZ length and growth rate (between 11/12 dps) using the GWAPP software. Due to the fact that a significant SNP identified by GWAS is frequently not the causal SNP—even when it is positioned into a gene—but rather indicates the genomic region in which the causal gene is placed [16], we mapped the genes located within a 10-Kb window (5 kb up and downstream) of the associated SNPs, as has previously been done [7,12]. The SNP with the highest score (−log10(*p*) = 7.44) was the SNP 1174196, found in the GWAS performed with the PR length at 4 dps. Additionally, we detected that this SNP has a high score for the time course growth at 5 and 6 dps: −log10(*p*): 6.96 and 5.38, respectively. The SNP 1174196 is mapped to the coding region (exon 6) of the AT3G04420.1 gene that encodes a transcription factor of the NAC domain-containing family (NAC048) and additional genes close to this SNP (within the 10 Kb window) are represented in Table 1. The second most significant SNP was the SNP 11031439, which had a score of −log10(*p*) = 6.777 and was obtained through the GWAS carried out using the PR length at 12 dps. SNP 11031439 was one of the SNPs with the highest score for the time course at 6, 8, 9, 10 and 11 dps (−log10(*p*) = 5.22, 5.94, 6.17, 6.31, 6.75, respectively). The nearest coding genes to this SNP are a Transmembrane protein (AT3G29034.1) and another NAC domain-containing protein 3 (AT3G29035.1). Other statistically significant SNPs found in the GWAS carried out at 2 and 3 dps are indicated in Table 1.

Although we did not find any significant SNPs using either the growth rate or the meristem length data, the maximum score for the growth rate and for the meristem length were 5.24, corresponding to SNP 6519608, and 4.95 corresponding to SNP 15777913, which mapped inside the second exon of a hypothetical protein (AT3G43970.1; see Table 1). Regarding EZ length, we detected a significant SNP (−log10(*p*) = 6.65) at position 2557492 that mapped close to a single gene (AT4G04990.1) that encodes a Serine/arginine repetitive matrix-like protein.

The spatial expression of the genes found in our GWAS was visualized using the Plant eFP browser Tissue Specific Root (Appendix A). We noticed that all the genes were expressed in the PR, except for two: the NAC domain-containing protein 49 (AT3G04430.1) and an F-box/associated interaction domain protein (AT3G18905.1). Interestingly, from the six genes highly expressed both in the meristem and in the elongation zones, three were identified in GWAS using the meristem length (AT3G43960.1, Cysteine proteinases superfamily protein; AT3G43970.1, a hypothetical protein; and AT3G43980.1, a Ribosomal protein from the S14p/S29e family protein). This was also the case for the unique gene that was found using the EZ data (AT4G04990.1, a Serine/arginine repetitive matrix-like protein DUF761) (Appendix A). Furthermore, there are three other genes that are also highly expressed in the meristematic and elongation zones that could be good candidates to evaluate their function in PR growth: AT3G62580.1, AT3G62600.1 and AT3G29035 which encode proteins from the Late Embryogenesis Abundant protein (LEA) family, DNAJ heat shock family and NAC3, respectively (Appendix A). As there are many cases where the expression of the gene reflects the place where the protein functions, all these data reinforce the hypothesis that these genes could be important for meristem and elongation length affecting PR growth.

For further analyses we selected only six out of the twenty-three genes found in our GWAS according to the trait, its maximum score, root expression profile and sequence variability between contrasting accessions. *NAC3* and *NAC048* were chosen for the PR length analysis, AT4G04990 a Serine/arginine repetitive matrix-like protein (DUF761), AT3G18900 a Ternary complex factor MIP1 leucine-zipper protein and AT3G43970, a hypothetical protein for the EZ length, growth rate and meristem length, respectively.

To further analyzed these genes, we first look for the SNPs using only the accessions that were grouped in the extremes of the category (short–large, slow–fast traits) that also, have available sequences in the 1001 Genomes Project. We found that all the four transcription factors (TFs) obtained in the GWAS analysis, belong to the NAC family and one of them *NAC048* has different SNPs in its exons depending on whether the accessions have a short or a large PR. For instance, at position 456 in *NAC048*, 27.7% of the accessions (Baa-1, Chat-1, Gie-0, C-24 and Sq-1) have a missense variant gaC/gaA that produces a change in the acidic amino acid residue (D/E), whereas only one accession with large PR length bares the same SNP. Additionally, at position 452 of the third exon of this gene, there is another missense variant (gGt/gAt) that leads to an amino acid change G/D in 16.6% of the accessions analyzed with a short PR (Baa-1, Chat-1 and Sq-1), while only 8.3% of the accessions with large PR length have this variant (Figure 4, Table 2). Moreover, in the fifth exon of *NAC048*, 33.3% of the accessions with large PR (An-1, Mrk-1, Ra-0, RRS-7) have a Gtg/Ctg change (V/L) and none of the accessions with short PR length have the same SNP (Figure 4, Table 2).

For *NAC3*, 33.3% of accessions with a short PR have a missense variant (Ttt/Gtt, F/V) in the third exon, compared with 9% of accessions with a large PR (Table 2). Conversely, the *TERNARY COMPLEX FACTOR* gene displays a missense variant (Gga/Cga, G/R) in 55.5% of the accessions with a fast growth rate, in comparison with 21.4% of accessions with a low growth rate. In contrast, when analyzing a missense variant in a cysteine proteinase superfamily protein, the proportion between accessions carrying a small and large EZ length is similar (Table 2). Furthermore, we found only synonymous variants in the coding regions of the *FATTY ACID DESATURASE 2* gene and in one gene that encodes for a hypothetical protein (AT3G43970).

These results suggest that *NAC048*, *NAC3* and the gene that encodes the ternary complex factor MIP1 leucine-zipper protein could be good candidates for regulators of primary root development among Arabidopsis accessions. To further look for evolutionary trends using a gene tree reconstruction approach, we analyzed only 18 of the 23 genes, because they were the only ones of which we could find sequence information from the Arabidopsis 1001 Genomes.

### 2.3. Phylogenetic Tree Reconstructions Reveal Additional Accessions for Further Studies

To investigate if there were genetic variants associated with the three phenotypic classes into which all the accessions used were divided for the four traits analyzed (short, medium and large/slow, medium and fast), we retrieved the available homologous sequences of the 18 genes with the highest scores in the GWAS (see Table 1). After alignment and a Maximum Likelihood reconstruction of phylogenetic gene trees, we did not find simple patterns of genotype-phenotype associations, i.e., in the case of PR length at 12 dps, accessions with short roots did not necessarily share the same mutations or indels, nor did the accessions with long PR. This finding is consistent with our analyses regarding the distribution of genetic variants among the phenotypic classes identified (see the previous section), as well as with the complex genetic basis of traits with quantitative variation. Nevertheless, we found that six out of eighteen genes had genetic variants and, in some cases, deletions common to several accessions (*FAD2* (AT3G12120) and *NAC3* (AT3G29035)) that warranted further analysis. Thus, we analyzed the nucleotide substitution patterns of *NAC3* (AT3G29035), Serine/arginine repetitive matrix-like protein DUF761 (AT4G04990), hypothetical protein (AT3G43970), Ternary complex factor MIP1 leucine-zipper protein (AT3G18900), *NAC048* (AT3G04420) and *FAD2* (AT3G12120) sequences and included their geographic origin into additional phylogenetic trees to investigate if the accessions with a particular root phenotype shared a common origin. We did not find direct associations between the country of origin and particular phenotypic classes, although some interesting groups were recovered. In general, most clades within a particular gene tree bore accessions with both small/slow and large/fast phenotypic classes, as well as medium-sized accessions. In the hypothetical protein (AT3G43970) we recovered the largest groups comprising of accessions with either small and medium meristem size or large/medium meristem size (Figure 5). Accessions that fall in the small classes are GA-0, ROU-0, SI-0, LP2-2 and GOT7, and for large classes we found PHW-2, MIR-0, Pu2-7, WA-1, PRO-0, BUR-0 and HS-0. Despite this general trend, within smaller clades, we did recover groups where two or more accessions that fell within the same phenotypic class were associated together (Figure 5, external green and orange lines).

Another trend was the recovery of large clades that shared common deletions. This was the case for FAD2 (AT3G12120; see Appendix A) but was particularly striking in *NAC3* (AT3G29035), where two clades were supported by a large 5′ and a small 3′ deletion, respectively. In the former clade, accessions from all three phenotypic classes were included, while in the latter clade only accessions with average or short PR length were associated (See Figure 5—NAC3). Upon an inspection of smaller clades across the six aforementioned genes, we found some accessions with particular characteristics and consistent associations across different gene trees. For instance, PRO-0 from Spain (scored with small PR length, small EZ and a slow growth rate), was consistently grouped as sister to accessions in the “large” phenotypic class (Appendix A). In particular, it was associated with genes such as *NAC3* (AT3G29035) and Ternary complex factor MIP1 leucine-zipper protein (AT3G18900) from SG-1 accession from Germany. SG-1 has a large PR length and a fast growth rate, while it was scored as medium-sized for EZ and meristem size. Another interesting accession is LP2-2 from the Czech Republic, which was also in the “small” or “slow” phenotypic class (PR length small; EZ small; and slow growth rate) in five out six genes, except for *FAD2* (AT3G12120), where its PR length was scored as medium (See Appendix A for PR gene expression).

### 2.4. Complex Patterns of SNPs and Indels in NAC3 and NAC048

Finally, for *NAC3* and *NAC048* we used the alignment of 79 homologs retrieved from various accessions to pursue the SNPs, and plotted the distribution of SNPs and number of alleles per locus (Figure 5). In *NAC3*, we focused on exon 3—spanning the region between 701 and 1470 pb in our alignment—where we found three peaks of high-density SNPs, two of them over a 2 × 10^−4^ score. In this exon, we counted 92 SNPs with two alleles and 2-point mutations with three alleles in positions 1080 and 1082 of our alignment, respectively (Appendix A). Contrasting these results with our phylogenetic tree for NAC3, the largest clade comprising accessions with a “small” PR display particular SNPs. In exon 3, accessions LU-1 (18 SNPs), NO-0 (2 SNPs), BAY-0 (1 SNP) and SQ-1 (3 SNPs) shared SNPs at positions 734 (C/A) and/or 1061 (T/G) of our alignment (Appendix A). Interestingly, these accessions are part of a clade that shares a deletion in the 3′ of the gene, spanning from position 4780 to 4880 in our alignment. In addition, in the other “small” clade, accessions ZDR-1 (1 SNPs) and MS-0 (2 SNPs) have a SNP in position 734 (C/A) of our alignment. In a clade comprising “large” PR length at 12 dps, accessions POG-0 (2 SNPs), RRS7.Salk (1 SNP) and PRO-0 (3 SNPs) have a common SNP (G/A) in exon 3 at position 834 of the alignment. Other notable SNPs are present in the 5′ section of the gene at positions 230 (A/C) and 242 (T/C), while an additional SNP in the 3′ section is in position 4559 (A/G) in our alignment.

In the case of *NAC048*, we focused on exon 3 (between 578 and 718 pb) and exon 5 (between 1080 and 1157 pb), where we previously found variability in our GWAS. Near to these two exons we found three peaks of high SNPs density (over 0.00125 score). In exon 3, we detected three SNPs with two alleles in positions 609, 618 and 659, while in exon 5 we identified two SNPs in positions 1105 and 1040 of our alignment with two alleles (Appendix A). In the phylogenetic tree for this gene, the clade which concentrates the accessions with a “small” PR phenotype corresponds to BAA 1, SQ-1 and CHAT-1. The latter had two SNPs in exon 3, at positions 618 (G/A) and 622pb (C/A) in our alignment, while in exon 5 these accessions did not bear any SNPs. The accessions with SNPs in exon 5 were BUR-0 and OLD-1 in position 1105 (G/A). These two accessions are near in our phylogenetic tree, but only BUR-1 had a large PR length. The other SNP in exon 5 was in position 1140 (G/C), and only seven accessions had it, but displayed a “medium” PR length phenotype.

## 3. Discussion

### 3.1. Natural Populations of Arabidopsis Have Wide Variation of Primary Root Length, Growth Rate and Features at the Cellular Level

In this work, we were able to characterize the enormous phenotypic variation in different morphological traits such as the primary root length, which varies from 3.14 cm to 9.5 cm at 12 dps in 124 Arabidopsis accessions as previously reported [17]. Furthermore, to explore the genetic basis of this root natural variation, we divided the accessions into three classes (short, medium and long) for PR, meristem and EZ length, and slow, intermediate and fast for growth rate. With this approach, we were able to compare more than 15 accessions in the contrasting groups (small/large) for PR and meristem length, in a different way to what has been reported under stress conditions that do not divide the accessions [10,11,12]. However, since our classification was made only with the mean value of each trait, without considering the standard deviation, some accessions could not be adequately classified. Nevertheless, this methodology allowed us to identify contrasting accessions that could be used for further analyses.

It has been shown that the length of both the meristem and elongation zone and the length of the fully elongated cells, establish the PR length [6,18,20,21,24]. Furthermore, a very high correlation has been found between the meristem length and the fully elongated cell length [18]. Interestingly, we found a medium, positive correlation between PR growth and meristem length and a positive and strong correlation between primary root growth rate and PR length that has not been addressed before. Furthermore, a short PR can be due only to a shorter meristem, without changes in the length of the fully elongated cortical cells [30,31], suggesting that different combinations of cell proliferation, elongation and differentiation account for overall PR growth. In our analysis, we noticed a moderate (as defined by [34]) positive linear correlation between EZ length with both the PR length and the meristem length, in contrast to what has been published before [18]. Therefore, we propose that the EZ length could also be an important trait to understand the length of the PR.

### 3.2. GWAS Uncover Novel Genes with the Potential to Explain the Natural Variation in the Length of the Primary Root, the Meristem, and the Elongation Zone

Remarkably, we found that five out of the seven SNPs identified in the different GWAS were statistically significant for the PR length over a time course growth for 2, 4, 5, 6, 8–12 dps, and this could be due to the minor allele frequency (MAF) utilized and/or the long-time course growth of 2 to 12 days. We used a MAF ≤ 0.05, which is used to eliminate SNPs with a low minor-allele frequency and to provide a better allele accuracy [35,36]. Besides, a prolonged course growth is important as two different studies have shown that the associations with some SNPs increase while others decrease, depending on the day when a particular trait is analyzed during root development [12,37]. Accordingly, we found that the *p*-Value of the SNP 11031439 increased from −log10(*p*) = 5.225 to 6.777 from day 6 to the end of the growth time course on day 12. This could be important to explain the medium correlation we found between PR length at day 5 and day 12, as the SNPs reflect changes in primary root growth. In addition, a correlation has been found between root growth due to physical constraints at 28 days and SNP scores [37].

Moreover, we used a 10-kb window for the most significant SNPs and discovered 23 candidate genes associated with the four traits evaluated. Only one of these, *FAD2*, an endoplasmic reticulum localized desaturase, has been described as a promoter of PR length, since the loss of function mutant shows a shorter PR compared to WT plants [33]. In addition, the overexpression of another of these genes, *NAC048*, results in a disorganized root vascular phenotype [38]. However, we could not find the participation of the remaining genes in PR growth but most of them (21 of the 23 genes found in GWAS), are expressed in different zones of the PR (Appendix A) [38,39,40,41,42].

Two of the genes we found have missense variants in their coding regions for accessions with short and long PRs, and belong to the NAC transcription factor family that has 117 members in Arabidopsis and plays important roles in plant development and stress response [43]. In addition, a gene that belong to the NAC domain transcriptional regulator superfamily (AT1G60340) and *NAC24* (AT1G60350) were found in a GWAS using the meristem length, and NAC6 was detected in a statistical epistatic interaction with two linked SNPs [17], suggesting the participation of this family in PR growth.

We found no significant SNPs in GWAS based on meristem length and growth rate. However, it has been reported that even low scores render causal SNPs. For instance, Slovak et al. (2020) [16] performed a GWAS on root growth rate and focused on its top association (−log10(*p*) = 4.7) and found that the *ARABIDOPSIS ADENYLATE KINASE 6* (AAK6) is a gene involved in root growth rate determination. In our study, we detected a gene that encodes a ternary complex factor MIP1 leucine-zipper protein (AT3G18900) that has a missense variant in 55% of accessions with a fast growth rate compared to those with a slow growth rate. Moreover, Deolu-Ajayi et al. [8] employed the root growth rate as an input for a GWAS study and identified significant SNPs different to those found here; some of the differences between their study and ours are that they used 6 to 9 days after germination (dag), compared to the 11 to 12 dps that we used for the quantification of this trait. Finally, we could identify a significant SNP that mapped near a Serine/arginine repetitive matrix-like protein (DUF761) in the GWAS using the EZ length, in contrast with Meijon and collaborators [18] who did not identify a statistically significant SNP.

### 3.3. Phylogenetic Gene Tree Reconstruction Allowed for the Investigation of Evolutionary Trends and Discovery of Additional Accessions for Further Studies

The fact that the associations between phenotypic classes and particular genetic variants are not straight forward, nor is the association between genetic variants and their geographic origin, is unsurprising, given the evolution of Arabidopsis, in particular its recent history. After the last glacial period there was a fast range expansion from Asian relict populations into the West, which translated into several extant populations with little genetic variability among them, as well as instances with long distance gene flow [44]. This can help explain counter-intuitive groupings as those documented here. A noteworthy result from our GWAS is that, except for one gene, the Serine/arginine repetitive matrix-like protein DUF761 (AT4G04990), all other genes map to chromosome 3, where other authors had previously documented the presence of genes related with root development [45]. In later studies, several accessions have been documented to harbor a SNP on this chromosome related to coding gene variants involved in different developmental processes, including root development [44]. Furthermore, a recent study that analyzed root architecture and developmental patterns in multiple accessions of Arabidopsis and also performed a GWAS approach, mapped a majority of genes to chromosome 3 [37]. In the case of NAC3, the SNPs we found in exon 3 could be a factor contributing to variability in PR length; it would be interesting to see if the variability found in this gene’s third exon is related with a particular lineage of Arabidopsis, or if it is a shared trait among different species within this genus. Towards this end, a phylogenetic analysis that includes closely related species, such as *A. lyrata* among others, would be informative. To further evaluate the functional consequences of the indels and SNPs in the resulting NAC3 protein, in silico protein modeling could allow for the exploration of potential changes in functional domains, while an experimental analysis of the accessions described above and a loss of function mutant for *NAC3* could also be of interest.

In addition, accessions with contrasting nucleotide size for this gene could be further analyzed in accession that have either, longest nucleotide sequence like C24 and Col-0 or a smaller one likeLP2-2. Overall size variation in the predicted gene sequence was also observed for *NAC048*, where accessions IS-0 and Col-0 have the longest gene, while BAY-0 has the shortest sequence. We stress that none of the identified genes in the study by LaRue and collaborators [37] coincide with ours, which again, could relate to the fact that they analyzed a different set of traits and did so in older plants. Thus, comparisons among GWAS studies for the same organ should pay attention to the temporal window when experiments were conducted in order to assess how comparable they can be.

## 4. Materials and Methods

### 4.1. Plant Materials and Growth Conditions

We used 124 Arabidopsis accessions obtained from 24 countries (http://gwas.gmi.oeaw.ac.at/ (accessed on 24 June 2022) (Appendix A). Seeds were sterilized by shaking them for five minutes in 100% ethanol, then transferred to a solution of chlorine 5%/SDS 1% for 13 min and washed four times in sterile water before sowing. Seeds were stratified for five days at 4 °C in darkness and then placed on square Petri dishes with 0.2X MS salts, 1% sucrose, 0.05% MES and 1% agar at pH 5.6 [46]. Plants were grown for five days post sowing (dps) under long-day conditions (16 h light/8 h dark) in growth chambers at 22 °C and then transferred to new vertical plates for seven more days under the same growing conditions. The time course growth of the primary root was performed during 12 dps, marking the position of the root tip every 24 h on the back of the plate (Appendix A). At the end of the growth time course, the plates were digitized at 300 dpi and the roots were measured using the Image J version Fiji software (NIH, USA) (using a ruler as a scale. Ten seedlings in two square Petri dishes (five on each plate) per accession were used for the time course growth. Every day the plates were moved randomly in the growth chamber to uniformize growth conditions. Furthermore, the 124 accessions were grown in seven different batches (13–24 accessions per batch) with Col-0 as internal control.

### 4.2. Pseudo-Schiff Assay

At 5 dps, ten seedlings per accession were fixed in 50% ethanol and 10% acetic acid at 4 °C for 24 h and were used to measure the root meristem and the elongation zone length. After fixation, the seedlings were incubated for 40 min in 1% periodic acid at 37 °C and washed three times with distilled water. Then, seedlings were incubated in 100 mM sodium metabisulfite, 0.15 N hydrochloric acid and 30 mg/mL propidium iodide solution for 15 min. They were washed three times with distilled water and placed in 2% DMSO/30% Glycerol for two days. Finally, the roots (10 for each accession) were cleared with a sodium iodide solution (5.6 M NaI, 8 mM Na_2_S_2_O_3_, 65% glycerol and 2% DMSO) over the slide to be observed under an optic or confocal microscope [29].

### 4.3. Microscopy Visualization

The equivalent tissue from ten seedlings from each accession was visualized using Nomarski optics using an Olympus BX60 microscope with a dry 40× objective and photographed with an Evolution MP COLOR camera of Media Cybernetics [29]. Confocal images were acquired using an inverted Nikon A1R+ STORM with a dry 20× objective.

### 4.4. Quantitative Analysis

At 5 dps, ten seedlings from each of the 124 accessions were used for quantification at the cellular level (Appendix A). In this work, the meristem size was measured from the Quiescent Centre to the cortical cell before the cell whose size is twice the size of the previous cell (onset of cell expansion). The size of the EZ was measured from the onset of cell expansion to the first bulge root hair (Figure 1c), also in the cortex tissue. Both observations were made in a median longitudinal optical section and the meristem and EZ length measures were done using the Image J version Fiji software (NIH, USA). The growth rate was obtained by subtracting the PR length of day 12 minus day 11.

### 4.5. Plant Phenotype Studies

The association between the PR length, the growth rate, the meristem or the EZ length was tested by Pearson’s correlation. The PR length during the 12 days of the time course, the PR growth rate at 11–12 days, the meristem and the EZ length were classified into small, medium and large sizes according to Sturge’s rule, which is used to fix the number of classes given the total number of observations (Appendix A). In this case, only three bins were specified employing the programming language R (R Core Team, 2022).

We performed histograms representing the density distribution of each of the four traits and checked the normality using the Kolmogorov-Smirnov test (Appendix A). The non-normal data (PR length 2–7 dps) were transformed into a normal distribution using the Box–Cox transformation to execute the GWAS using a mixed model algorithm, the GWAPP software (Transplant project, Vienna, Austria).

### 4.6. Genome-Wide Association Studies

Genome-wide association mapping was performed using the GWAPP web interface, which contains genotypic information for −206,000 SNP markers [47] (http://gwas.gmi.oeaw.ac.at/ (accessed on 24 June 2022). The 124 accessions used in this work are part of the 1386 publicly available Arabidopsis accessions that have been sequenced and used in GWAPP (Transplant project, Vienna, Austria). GWAS were conducted with the accelerated mixed-mode (AMM) to identify associations between the traits of the 124 accessions and the SNPs available in the database. Manhattan plots were retrieved from GWAPP, filtering out a minor allele frequency (MAF) ≥ 0.05 [8,35]. These plots represent the genomic position of each SNP and its association [−log10(*p*-Value)]. The significance of SNP associations was determined at the 5% FDR threshold computed by the Benjamini–Hochberg–Yekutieli method to correct for multiple testing [48]. For each of the candidate genes, the annotations were retrieved from the AraGWAS Catalog (https://aragwas.1001genomes.org/ (accessed on 24 June 2022) and TAIR10 (www.arabidopsis.org/ (accessed on 24 June 2022).

### 4.7. Polymorphism Patterns in the Selected Genes in Accessions with Extreme Phenotypes

Sequence data from the 1001 genome project [44] (http://signal.salk.edu/atg1001/3.0/gebrowser.php (accessed on 24 June 2022) were used to analyze polymorphism among accessions with contrasting phenotypes (accessions with small and large PR, meristem, elongation zone length and fast and slow growth rate). SNPs located in the 10-kb window (5 kb up and downstream) of the associated SNPs were mapped and examined for natural variation [35]. SNPs information of all available accessions was compiled and contrasted. In order to determine the effects of these variations and their exact positions, Variant Effect Predictor (VEP; [49]) was executed, with default parameters, on all SNPs found in these windows (https://plants.ensembl.org/Arabidopsis_thaliana/Tools/VEP (accessed on 12 July 2022). Those SNPs found in the greatest number of significant accessions for each trait were counted to find SNPs with biological relevance.

### 4.8. Phylogenetic Tree Construction and Analysis

We extracted the genomic sequence of 18 genes with the highest scores in the GWAS analyses (see Appendix A in bold letters) for 76 of the 124 accessions with available genomes (Appendix A indicated in blue). We retrieved the homologs from Arabidopsis 1001 Genomes—Genome Express Browser 3.0 (http://signal.salk.edu/atg1001/3.0/gebrowser.php, accessed on 2 August 2022) and removed incorrect symbols in sequences to maintain an appropriate FASTA file for each gene. A total of 1368 nucleotide sequences (18 genes × 76 homologs) were analyzed. FASTA files for each of the 18 genes were aligned in MAFFT software (https://mafft.cbrc.jp/alignment/server/, accessed on 6 August 2022) using the default settings with 1.53 of gaps opening penalty [50,51].

Phylogenetic analyses were performed with IQ-tree software (http://www.iqtree.org/, accessed on 10 August 2022) [52,53]. First, we obtained the appropriate substitution model based on the Bayesian criterion using each gene alignment and then got the inference tree for each gene. For the 18 genes, we constructed a Maximum Likelihood (ML) phylogenetic tree with 1000 bootstraps in ultra-fast analysis to support the branches. For 6 out of 18 genes which displayed more sequence heterogeneity among accessions (AT3G04420, AT3G29035, AT4G04990, AT3G18900 and AT3G43970), we conducted further analyses. To increase the resolution of the phylogenetic tree, we performed a standard ML analysis with 100 bootstraps. For each phylogenetic tree, we obtained a consensus tree and we used the Newick tree format in iTOL V4 (https://itol.embl.de/, accessed on 15 August 2022) to visualize and include the branch colors and tags of the country of origin [54].

Single nucleotide polymorphisms were extracted with a tool set Adegenet 1.3.1, R package [55] (31 August 2022). We analyzed the nucleotide sequences extracted from the Arabidopsis 1001 Genomes database and we determined the exon positions through a BLAST(NCBI) analysis (Appendix A). Our analyses were based on the alignments and we generated a matrix with the nucleotide punctual changes for each of the 6 genes that were studied in detail.

### 4.9. In Silico Analysis

The in silico gene expression analysis in the roots was visualized using the ePlant browser of Araport, Tissue-Specific Root eFP (http://bar.utoronto.ca/eplant/ (accessed on 24 June 2022)).

## 5. Conclusions

In this work, we use a combined methodology of GWAS and phylogenetic gene tree reconstruction in order to search for candidate genes and further analyze relevant genetic variants that participate in PR development. The GWAS allowed us to identify 23 genes that could participate in the control of proliferation and elongation in the PR. However, the alignments of six of these genes and the reconstruction of phylogenetic trees did not show robust patterns in genotype–phenotype associations. Interestingly, two members of the NAC family, *NAC048* and *NAC3*, display different SNPs in their exons in contrasting accessions of PR length and, in some cases, these accessions group together, according to our phylogenetic analysis. These genes can be good candidates to validate their function in PR growth through loss or gain of function mutants.

## Figures and Tables

**Figure 1 plants-11-03162-f001:**
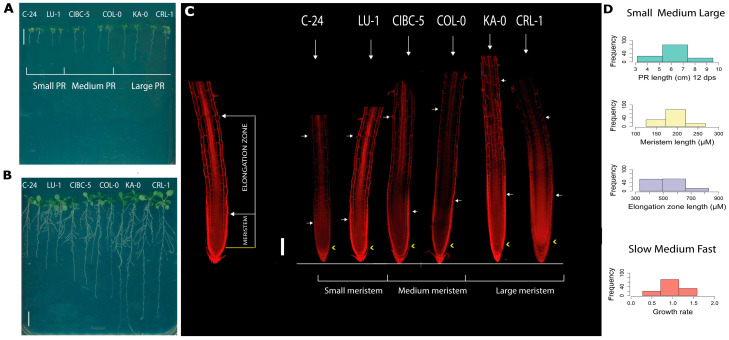
Representative accessions (two seedlings per accession) with short, medium and large PR at 5 dps (**A**) and 12 dps (**B**). (**C**) Median longitudinal confocal images of PR roots at 5 dps observed in (**A**); the meristem and the elongation zones are shown. The yellow arrow marks the QC and the initiation of the meristem and the white arrows indicate the end of the meristem and the initiation of the EZ that ends with the first epidermal cell with a bulge hair root. Roots were stained with propidium iodide. The six accessions: C-24, LU-1, CIBC-5, COL-0, KA-0 and CRLl-1 have the same classification regarding their PR and meristem length. (**D**) For each of the 124 accessions (*n* = 10 seedlings used per accession), we plotted the frequency of small, medium and large regarding primary root length and root growth rate at 11 and 12 dps and the meristem and elongation zone length of 5 dps roots using Sturge’s rule to see the distribution of the values. In (**A**,**B**) scale bar = 1 cm, in (**C**) the bar represents 100 μM.

**Figure 2 plants-11-03162-f002:**
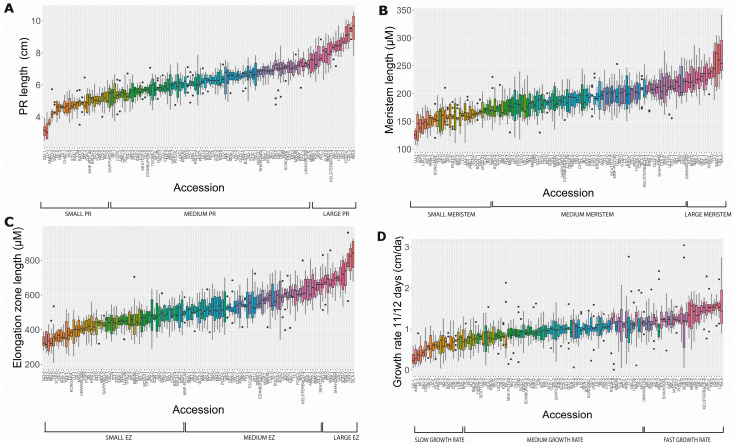
Natural variation of PR traits in the 124 accessions used in this study (*n* = 10 seedlings used per accession). The box plot represents the distribution of the PR length at 12 dps (**A**), the meristem (**B**) and elongation zone length (**C**) into small, medium and large. The growth rate from 11 to 12 dps of the accessions is indicated as slow, medium and fast (**D**). The boxes show the first and third quartiles, the vertical line, the median and the whiskers go from each quartile to the maximum or minimum value.

**Figure 3 plants-11-03162-f003:**
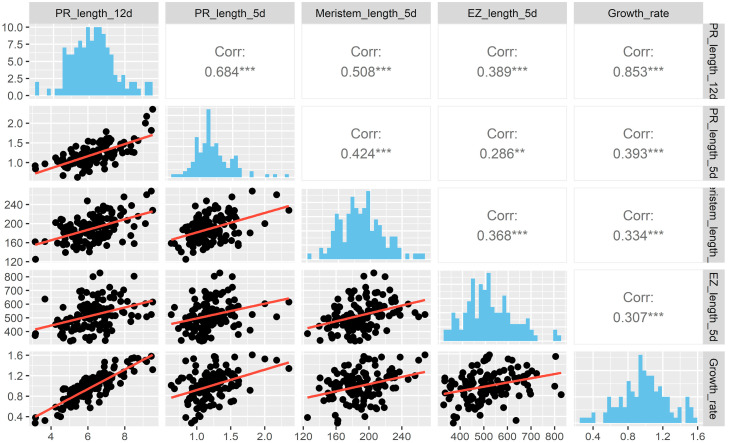
Pair plots between the four PR traits used in this study. The upper panel shows the correlation between the traits, the lower panel, the scatter plots of the traits and on the diagonal, the histograms. PR length was analyzed at 5 and 12 dps. Significant value is indicated (***) *p* < 0.001.

**Figure 4 plants-11-03162-f004:**
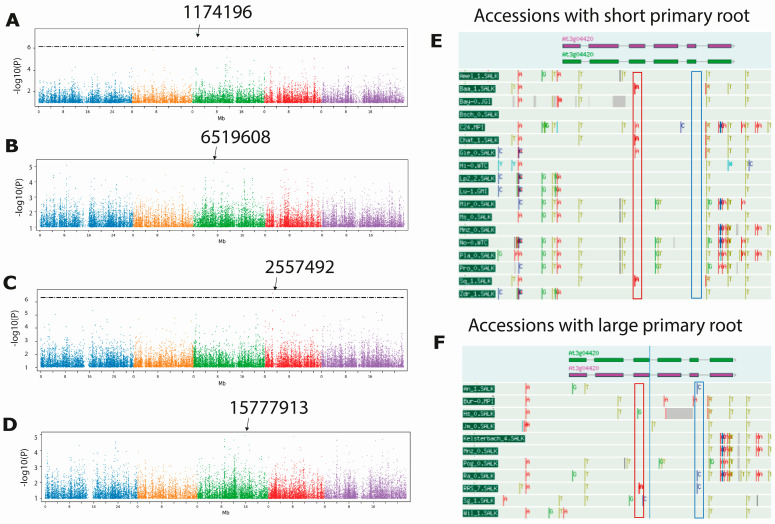
GWAS were performed using the median of each of the traits analyzed and significant associations were identified using an accelerated mixed model. The threshold is plotted as a dotted horizontal line. (**A**) PR length at 12 dps, (**B**) growth rate at 11/12 dps, (**C**) elongation zone and (**D**) meristem length. The position of the SNPs with high scores is indicated in the upper part of the Figure (**A**–**D**). (**E**) SNPs found in *NAC048* homologs from accessions with short PR length. The SNP gGt/gAt in the third exon of *NAC048* is found in 16% of accessions (Baa-1, Chat-1 and Sq-1), whereas the gaC/gaA change in the fifth is not present, (**F**) SNPs observed in *NAC048* from accessions with large PR length, the Mrk-0 is not shown in the image since it is not found in the 1001 Genomes project page. The SNP gGt/gAt in the third exon of *NAC048* is found in 8.3% (RRS-7) of accessions, whereas the gaC/gaA change in the fifth is detected in 33.3% of accessions (An-1, Mrk-1, Ra-0, RRS-7) with large PR.

**Figure 5 plants-11-03162-f005:**
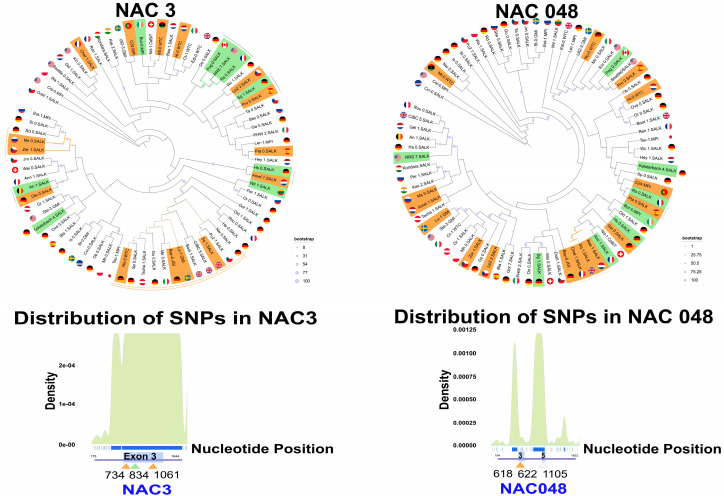
Phylogenetic trees of *NAC3* and *NAC048* and SNPs distribution derived from the alignment of homologs from 79 accessions. (**A**) *NAC3* and *NAC048* phylogenetic trees. Accessions with small PR are colored in orange; accessions with large PR are green. External green line: a clade with accessions with large PR; External orange line: a clade with small PR; External orange double line: a clade with deletion in the 3′ of *NAC3*; External gray line: clades that comprise accessions with contrasting character states (small, large). (**B**) Distribution of SNPs density across *NAC3* and *NAC048*. Bottom blue line: consensus gene sequence. Blue boxes: Exon regions. Orange triangles: SNPs associated with small PR clades. Green triangles: SNPs associated with large PR clades. White triangle: SNPs not associated with a large or small PR phenotype.

**Table 1 plants-11-03162-t001:** List of the 23 Genes Identified by GWAS within a 10 kb window.

Chr	SNP Position	Maximum Score	MAF	Genes 10 Kb Window	Trait
3	1174196	7.444 *	0.235	AT3G04420.1 NAC domain-containing protein 48 (NAC048) AT3G04410.1 NAC domain transcriptional regulator superfamily protein AT3G04430.1 NAC domain-containing protein 49 AT3G04440.1 Plasma-membrane choline transporter family protein AT3G04443.1 hypothetical protein	PR length 4 dps PR length 5 dps PR length 6 dps
3	3866768	6.815 *	0.056	AT3G12120.1 fatty acid desaturase 2 (FAD2) AT3G12130.1 KH domain-containing protein/zinc finger (CCCH type) family protein. AT3G12140.1 Emsy N Terminus (ENT)/plant Tudor-like domain-containing protein	PR length 2 dps
3	11031439	6.777 *	0.073	AT3G29034.1 Transmembrane protein AT3G29035.1 NAC domain-containing protein 3	PR length 6 dps PR length 8 dps PR length 9 dps PR length 10 dps PR length 11 dps PR length 12 dps
4	2557492	6.650 *	0.057	AT4G04990.1 Serine/arginine repetitive matrix-like protein (DUF761)	EZ length
3	23146095	6.664 *	0.081	AT3G62570.1 Tetratricopeptide repeat (TPR)-like superfamily protein AT3G62580.1 Late embryogenesis abundant protein (LEA) family protein AT3G62590.1 alpha/beta-Hydrolases superfamily protein AT3G62600.1 DNAJ heat shock family protein	PR length 2 dps PR length 3 dps
3	6519608	5.246	0.317	AT3G18900.1 Ternary complex factor MIP1 leucine-zipper protein AT3G18905.1 F-box/associated interaction domain protein AT3G18910.1 EIN2 targeting protein 2 AT3G18930.1 RING/U-box superfamily protein	Growth rate 12 dps
3	15777913	4.954	0.081	AT3G43960.1 Cysteine proteinases superfamily protein AT3G43970.1 Hypothetical protein AT3G43980.1 Ribosomal protein S14p/S29e family protein AT3G43990.1 Bromo-adjacent homology (BAH) domain-containing protein	Meristem length

* statistically significant.

**Table 2 plants-11-03162-t002:** SNPs found in the six candidate genes obtained by GWAS in contrasting accessions.

Gene	Trait	CDS Position	Nucleotide Change	Amino Acid Change	Trait Classification	Percentage of Accessions
AT3G04420.1 NAC domain containing protein 48	PR length 4 dps	452 Third exon	Missense variant gGt/gAt	G/D	Short	16.6%
					Large	8.3%
		814 Fifth exon	Missense variant Gtg/Ctg	V/L	Short	0%
					Large	33.3%
AT3G12120.1 Fatty acid desaturase 2	PR length 2 dps	1059 First exon	Synonymous variant gtA/gtG	V	Short	38%
					Large	45%
AT3G29035.1 NAC domain containing protein 3	PR length 12 dps	847 Third exón	Missense variant Ttt/Gtt	F/V	Short	33.3%
					Large	9%
AT3G43960.1 Cysteine proteinases superfamily protein	EZ	224 First exón	Missense variant gAt/gCt	D/A	Short	14.2%
					Large	17.6%
AT3G43970.1 hypothetical protein	Meristem length	45 Second exon	Synonymous variant gcT/gcG	A	Short	0%
					Large	50%
AT3G18900.1 ternary complex factor MIP1 leucine-zipper protein	Growth rate	87 Fourth exon	Missense variant Gga/Cga	G/R	Slow	21.4%
					Fast	55.5%

## Data Availability

Not applicable.

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
