# Peer review of "Combined Approach of GWAS and Phylogenetic Analyses to Identify New Candidate Genes That Participate in Arabidopsis thaliana Primary Root Development Using Cellular Measurements and Primary Root Length"

_plants, 2022, doi:10.3390/plants11223162_

Round 1

Reviewer 1 Report

The ability of plants to regulate their root elongation is essential to their survival in a changing environment. There is no doubt about the importance and relevance of this work. The identification of each new root growth regulator gene expands our ability to understand the mechanism of this process.

I have a few small remarks:

Line 69: "there are" is repeated two times;

Line 403: space needed between "response" and "[44];

Line 480: Na2S2O3 - numbers should be indexes.

Author Response

Comments and Suggestions for Authors

The ability of plants to regulate their root elongation is essential to their survival in a changing environment. There is no doubt about the importance and relevance of this work. The identification of each new root growth regulator gene expands our ability to understand the mechanism of this process.

The authors appreciate the positive opinion of the reviewer about the contribution of our work to the field and are very grateful for all the valuable corrections.

I have a few small remarks:

Line 69: "there are" is repeated two times;

It has been corrected (now line 70)

Line 403: space needed between "response" and "[44];

It has been corrected (now line 431 and cite [43].

Line 480: Na2S2O3 - numbers should be indexes.

It was fixed to Na2S2O3 (now line 511)

Reviewer 2 Report

1.     Line 160-167, how about the associate relationship among PR length of 5d, Meristem length 5d and EZ length 5d?

2.     Line 263-264, how to select the 18 genes from 23 genes?

3.     Line 487-495, how many seedlings were used to calculate the EZ or PR length for each accession at each time point ? Whether the trial was replicated?

4.     Line 509-511, can the SNPs from public database represent the genotypes of 124 accessions?

5.   Why divide the PR length, EZ length etc. into three categories, respectively, instead of directly conducting GWAS analysis between phenotype and genotype of the 124 accessions?

Author Response

Comments and Suggestions for Authors

  1. Line 160-167, how about the associate relationship among PR length of 5d, Meristem length 5d and EZ length 5d?

     Thank you for your suggestion, we have now included the PR length at 5 dps (Fig. 3) to understand the relation between the five measurements (PR length at 5, 12d, Meristem, EZ and growth rate). We rewrote the paragraph to analyze the correlation between these traits:

     “In order to understand how these cellular parameters and PR length are associated, we performed multiple correlations among the four variables (Figure 3), finding that the highest correlation (0.85) detected was between the PR length at 12 dps and the growth rate that was obtained using the differences in growth between day 11 and 12.  Whereas, the correlation between the PR length at 5 dps and 12 dps is 0.684, showing differences in growth during development. Different to what we expected, both the meristem and the EZ size have a higher significant correlation with the PR length at 12 days than at 5 days suggesting the importance of considering additional temporal measurements to explain the PR length among different accessions. Moreover, there was a low positive correspondence among the meristem size, the growth rate and the EZ length, in contrast to a negative correlation between the EZ and meristem length previously reported [18] (Figure 3).” (lines 170-180).

  1. Line 263-264, how to select the 18 genes from 23 genes?

The 18 genes were selected because we were able to find their sequence in the Arabidopsis 1001 Genomes - Genome Express Browser 3.0. We have rewritten this paragraph to be clearer.

“To further look for evolutionary trends using a gene tree reconstruction approach, we analyzed only 18 of the 23 genes because they were the only ones of which we can find sequence information from the Arabidopsis 1001 Genomes.” (lines 280-283)

  1. Line 487-495, how many seedlings were used to calculate the EZ or PR length for each accession at each time point? Whether the trial was replicated?

     Thank you for your observation. In the case of the EZ length measure, we used 10 seedlings to quantify this length and 10 more for PR course growth. We did not carry out replicates in EZ length as this number of seedlings is common for quantitative cellular measurements and for GWAS according to previous reports (Meijon et al., 2014). Moreover, the root measures of the 124 accessions were performed in 7 batches (13-24 accessions per batch) using Col-0 as internal control.  As we did not detect batch effects according to a one-way ANOVA analysis, we combine all the batches in a single GWAS per trait. We have added this information in the material and methods and in the results sections (lines 500- 502) and aggregated another supplementary Figure (Figure Supplementary S1).

      We have added this new sentence to the article:

  “The 124 accessions were grown in seven different batches with Col-0 as internal control and batch effect was not detected (Supplementary Figure S1). ” (lines 99- 100)

  1. Line 509-511, can the SNPs from public database represent the genotypes of 124 accessions?

     Yes, the 124 accessions are part of the 1386 publicly available A. thaliana accessions that have been sequenced and used for the GWAPP platform. We added this information to be clearer:

   “The 124 accessions used in this work are part of the 1386 publicly available Arabidopsis accessions that have been sequenced and used in GWAPP” (lines 544-545).

  1. Why divide the PR length, EZ length etc. into three categories, respectively, instead of directly conducting GWAS analysis between phenotype and genotype of the 124 accessions?

     Thank you for your observation, we used two separate methodologies to analyze our data. First, we divide the phenotype of the four traits into three categories in order to select contrasting accessions and use them to posteriorly detect polymorphism and associations in the phylogenetic tree.  Second, we performed the GWAS using the data from the 124 accessions to identify candidate genes related to PR development.

We added a paragraph to be clearer: “This classification allowed us to detect contrasting accessions to further analyze for particular polymorphisms in the genes obtained by GWAS as well as associations in the phylogenetic trees (see below sections 2.2 and 2.3).” (lines 130-132)

We also modified the sentence in lines 187-189 to clarify the idea: “We performed independent GWAS using measurements from the 124 accessions for PR course growth (2-12 dps), meristem size, EZ length and growth rate (between 11/12 dps) using the GWAPP software.
